# CTRP9 Mediates Protective Effects in Cardiomyocytes via AMPK- and Adiponectin Receptor-Mediated Induction of Anti-Oxidant Response

**DOI:** 10.3390/cells9051229

**Published:** 2020-05-15

**Authors:** Bernd Niemann, Ling Li, Dorothee Siegler, Benedikt H. Siegler, Fabienne Knapp, Jakob Hanna, Muhammad Aslam, Michael Kracht, Rainer Schulz, Susanne Rohrbach

**Affiliations:** 1Department of Cardiac and Vascular Surgery, Justus Liebig University Giessen, 35392 Giessen, Germany; bernd.niemann@chiru.med.uni-giessen.de; 2Institute of Physiology, Justus Liebig University Giessen, 35392 Giessen, Germany; Ling.Li@physiologie.med.uni-giessen.de (L.L.); dorothee.siegler@hotmail.com (D.S.); Benedikt.H.Siegler@chiru.med.uni-giessen.de (B.H.S.); Fabienne.Knapp@physiologie.med.uni-giessen.de (F.K.); hannaj@klilu.de (J.H.); Rainer.Schulz@physiologie.med.uni-giessen.de (R.S.); 3Department of Cardiology and Angiology, Justus Liebig University Giessen, 35392 Giessen, Germany; muhammad.aslam@physiologie.med.uni-giessen.de; 4Rudolf Buchheim Institute of Pharmacology, Justus Liebig University Giessen, 35392 Giessen, Germany; Michael.Kracht@pharma.med.uni-giessen.de

**Keywords:** heart failure, ROS, AMPK, C1q family, cardiomyocyte

## Abstract

The C1q/tumor necrosis factor-alpha-related protein 9 (CTRP9) has been reported to exert cardioprotective effects, but its role in the right ventricle (RV) remains unclear. To investigate the role of CTRP9 in RV hypertrophy and failure, we performed pulmonary artery banding in weanling rats to induce compensatory RV hypertrophy seven weeks after surgery and RV failure 22 weeks after surgery. CTRP9 expression, signal transduction and mechanisms involved in protective CTRP9 effects were analyzed in rat and human RV tissue and cardiac cells. We demonstrate that CTRP9 was induced during compensatory RV hypertrophy but almost lost at the stage of RV failure. RV but not left ventricular (LV) cardiomyocytes or RV endothelial cells demonstrated increased intracellular reactive oxygen species (ROS) and apoptosis activation at this stage. Exogenous CTRP9 induced AMP-activated protein kinase (AMPK)-dependent transcriptional activation of the anti-oxidant thioredoxin-1 (Trx1) and superoxide dismutase-2 (SOD2) and reduced phenylephrine-induced ROS. Combined knockdown of adiponectin receptor-1 (AdipoR1) and AdipoR2 or knockdown of calreticulin attenuated CTRP9-mediated anti-oxidant effects. Immunoprecipitation showed an interaction of AdipoR1 with AdipoR2 and the co-receptor T-cadherin, but no direct interaction with calreticulin. Thus, CTRP9 mediates cardioprotective effects through inhibition of ROS production induced by pro-hypertrophic agents via AMPK-mediated activation of anti-oxidant enzymes.

## 1. Introduction

The adipokine adiponectin has direct beneficial effects on cardiomyocytes in several pathological heart conditions, including ischemia-reperfusion injury or cardiac hypertrophy [1,2]. The adiponectin-mediated attenuation of cardiomyocyte hypertrophy involves activation of AMP-activated protein kinase (AMPK) and inhibition of extracellular-signal regulated kinase (ERK) signaling [1,2]. A family of structural and functional adiponectin paralogs, comprising 15 members so far, was discovered and designated as C1q/tumor necrosis factor-alpha-related proteins (CTRPs) [3,4]. Previous studies from our group have shown that the adiponectin paralogs CTRP2 and CTRP7, which were among the first described CTRPs, cannot sufficiently compensate for the age-associated loss of adiponectin [5,6]. Cardiac expression of CTRP9 exceeds adiponectin by more than 100-fold [7]. Furthermore, the heart was identified as an organ expressing highest CTRP9 tissue levels, even exceeding adipose tissue where CTRP9 was originally discovered [8,9]. CTRP9 is secreted by cardiac cells and may function locally as a so-called “cardiokine” through paracrine and autocrine effects [7,8,9]. The cardiac secretome influences the local microenvironment and is increasingly recognized as a major regulator of cardiac physiology and pathophysiology. CTRP9 has been reported to activate a number of different signaling pathways including AMPK, Akt, mitogen-activated protein kinases (MAPKs) and protein kinase A (PKA). Similar to the effects described for adiponectin (acrp30), CTRP9 acts as a potent cardioprotective molecule. CTRP9 controls whole body energy balance via central and peripheral mechanisms and protects against diet-induced obesity and type 2 diabetes [10,11]. It reduces myocardial infarct size and cardiomyocyte apoptosis following ischemia-reperfusion in mice, improves cardiac function in diabetic mice and attenuates adverse cardiac remodeling after myocardial infarction [7,12,13]. Recently, a similar protection against ischemia-reperfusion injury has also been reported in rats [14]. However, capillary endothelial cell-derived CTRP9 has also been shown to trigger cardiomyocyte hypertrophy in a paracrine manner and to promote cardiac failure in a mouse transverse aortic constriction (TAC)-model [8]. In addition, locally expressed mediators, including adiponectin or the related CTRPs, may represent novel biomarkers for the prediction of prognosis and progression of cardiovascular diseases [15,16,17,18]. Elevated CTRP 3 levels, for example, have been related to the prevalence and severity of coronary artery disease, suggesting its potential as a novel biomarker [19]. Similarly, it has been shown in a single-center prospective longitudinal study that adiponectin could be utilized as a biomarker for the prediction of restenosis and worse prognosis in patients with coronary artery disease [20].

Although there are various reports on CTRP9 in left ventricular (LV) diseases, there are no data related to right ventricular (RV) pathologies so far. However, the RV and the LV differ in many aspects: (a) The two chambers have distinct embryological origins: the mammalian LV originates from the classical primary or first heart field, while the RV originates from a distinct secondary heart field [21]. (b) Metabolic changes and signaling responses in the mechanically stressed RV display several unique features and show major differences to responses in LV hypertrophy [22]. (c) The mass of the crescent-shaped RV is approximately one-sixth of the cone-shaped LV, but RV volume is larger than LV volume. The unique anatomy, myocardial ultrastructure and coronary physiology of the RV reflect a high-volume, low-pressure pump unlike the LV [23]. (d) The relative increase in RV afterload in severe pulmonary hypertension surpasses several-fold the relative increase in LV afterload even in severe systemic hypertension. Thus, although many features of LV remodeling in response to pressure overload have been elucidated over the last years and decades, these cannot be simply extrapolated to the RV. Accordingly, the role of CTRP9 in RV remodeling is not yet understood. Therefore, the aims of the present study were (1) to investigate whether CTRPs might play a role in RV failure, (2) to analyze the impact of exogenous CTRP9 on RV cardiomyocytes, and (3) to identify the mechanisms and signal transduction involved in CTRP9-mediated effects in RV cardiomyocytes.

## 2. Materials and Methods

### 2.1. PAB Model

Pulmonary artery banding (PAB) was performed in 6 weeks old male hybrid rats (Wistar Kyoto × Lewis rats). In brief, an anterolateral thoracotomy was performed under isoflurane anesthesia in fully ventilated rats and a 1.1 mm Weck Horizon ^®^ Titanium Clip (Teleflex Medical GmbH, Fellbach, Germany) was placed around the pulmonary artery. Sham-operated animals served as age-matched controls. At the time of surgery the clip was not constricting. However, as the animals grow, the clip became constricting, resulting in a stage of compensatory RV hypertrophy seven weeks after surgery and a stage of beginning RV failure 20–22 weeks after surgery. Approximately 24–26 weeks after surgery up to 50% of the animals died from heart failure. Therefore, PAB animals and age-matched, sham-operated controls were killed seven and 22 weeks after surgery (n = 8 per group). Two-dimensional and M-mode echocardiographic examinations were performed in accordance with the criteria of the American Society of Echocardiography using a Vevo2100 system (FUJIFILM VisualSonics Inc., Amsterdam, The Netherlands), evaluating both cardiac geometry and function. The animals were also inspected for clinical markers of RV failure, including enlarged liver and ascites. All animal experiments were performed according to the regional authorities and ethics committees for animal research and complied with the directive 2010/63/EU of the European Parliament on the protection of animals used for scientific purposes. The experiments were registered under the number G14-2017.

### 2.2. Adult Human Tissue and Blood Samples

Right atrial (RA), epicardial adipose, mammary artery and blood samples were obtained from patients undergoing routine cardiac surgery. The use of human tissue was approved by local ethics committees (65/10, 229/18) and informed patient consent was obtained before surgery. Human umbilical vein endothelial cells (HUVEC) were isolated from umbilical cords obtained from the Department of Obstetrics and Gynecology of the University Hospital Giessen after approval from the local ethics committee (18/13) and informed consent from the patients. The use of human tissue conformed to the principles outlined in the Declaration of Helsinki.

### 2.3. ELISA

Plasma brain natriuretic peptide (BNP) concentrations were measured by using a commercial Enzyme-Linked-Immunosorbent-Assay (Rat BNP ELISA-kit, RayBio, Peachtree Corners, GA, USA) according to the manufacturer’s instructions.

### 2.4. Isolation of Recombinant CTRPs

Recombinant protein was produced by cloning full-length mouse CTRP9 into pENTR™/D-TOPO^®^ (ThermoFisher Scientific, Dreieich, Germany) and maintained in *Escherichia coli* strain TOP10. After recombination into pDEST17, the N-terminal His6-tagged fusion protein was produced in *E. coli* strain BL21-AI, isolated from the lysed bacterial pellet with a nickel-affinity column (Amocol Bioprocedures Limited, Teltow, Germany), eluted with imidazole-containing buffer, and dialyzed against phosphate buffered saline (PBS). Potential endotoxin contaminants were removed with the EndoTrap Red Kit (Hyglos GmbH, Bernried am Starnberger See, Germany). Absence of endotoxin was verified with the Pierce Chromogenic Endotoxin Quant Kit (ThermoFisher Scientific, Dreieich, Germany). The lower limit of the kit is 0.01 EU/mL.

### 2.5. Isolation of Recombinant CTRP9 from HEK293 Cells

Full length mouse CTRP9 containing an N-terminal hemagglutinin (HA)-Tag was also transferred into pcDNA 3.1 DEST by recombination. Transient transfections were performed in HEK-293T cells using Lipofectamine^®^ according to the manufacturer’s protocol (Invitrogen). Twenty-four hours after transfection, cell culture medium was replaced with serum-free medium supplemented with vitamin C (0.1mg/mL). Supernatants were collected three times, every 48 h, pooled, and purified using a µMACS HA Isolation Kit (Miltenyi Biotec, Bergisch Gladbach, Germany). The purified protein was dialyzed against 20 mmol/L HEPES (pH 8), containing 125 mmol/L NaCl.

### 2.6. Isolation of Adult Cardiomyocytes, Endothelial Cells and Fibroblasts

Adult rat ventricular cardiomyocytes (ARVCs) were isolated from rats as described in greater detail previously [24]. Briefly, hearts were excised under deep anesthesia, transferred rapidly to ice-cold saline, and mounted on the cannula of a Langendorff perfusion system. Hearts were perfused first for 10 min in a non-re-circulating manner with a calcium-free perfusion buffer, then for 20–25 min in a re-circulating manner in a buffer supplemented with collagenase and 25 µmol/L calcium. Thereafter, RV and LV were separated. Afterwards both ventricles were minced separately and incubated for another 5 min in re-circulating buffer. The remaining cell solution was filtered through a 200 µm nylon mesh. The suspension was centrifuged at 25× *g* for 10 min to pellet down the RV and LV cardiomyocytes and the supernatant contained mostly the endothelial cells (ECs) and fibroblasts (FBs). Cardiomyocytes were re-suspended in buffer with a stepwise increase in calcium and finally transferred to culture medium (M199 supplemented with carnitine (2 mmol/L), creatine (5 mmol/L), and taurine (5 mmol/L)). Rat cardiomyocytes were attached to culture dishes by pre-coating of the dishes with 4% (*v*/*v*) fetal calf serum (FCS).

The remaining cells (supernatant after centrifugation) were centrifuged at 250× *g* for 10 min and the pellet was resuspended in 1 mL of endothelial cell basal medium (PromoCell^®^, Heidelberg, Germany) and incubated with magnetic beads (Invitrogen) pre-coated with anti-rat CD31 (ThermoFisher Scientific) for 1 h at 4 °C with end-to-end rotation. The microvascular endothelial cells coupled to magnetic beads were separated with a magnet, washed with endothelial cell basal medium, and seeded on 35 mm culture dishes. This procedure removed over 95% of endothelial cells from the mixture and the rest of the cells were seeded as cardiac fibroblasts in M199 medium supplemented with 10% FCS. All animals were handled in accordance with the directive 2010/63/EU of the European Parliament on the protection of animals used for scientific purposes.

Phenylephrine (10 µmol/L), propranolol (1 µmol/L), angiotensin II (100 nmol/L), endothelin-1 (100 nmol/L), norepinephrine (1 µmol/L), isoproterenol (5 µmol/L), adenine 9-β-d-arabinofuranoside (Ara A, 500 µmol/L), SB 202190 (5 µmol/L), prazosin (1 µmol/L), U0126 (10 µmol/L), metformin (500 µmol/L), H_2_O_2_ (1 µmol/L), paraquat (50 µmol/L), PX 12 (10 µmol/L), and cyclosporine A (CsA, 1 µg/mL) were purchased from Sigma-Aldrich Chemie GmbH (Schnelldorf, Germany) and used at the indicated concentrations.

### 2.7. Isolation of Human Cardiomyocytes and HUVECs

Myocytes were enzymatically isolated from human RA trabeculae as described before by others [25]. In brief, tissue was cut into chunks of approximately 1 mm^3^ using a tissue chopper (McIlwain, Campden Instruments Ltd., Loughborough, UK). First, the chunks were incubated in low calcium buffer containing 4 U/mL Sigma type XXIV protease by gently shaking and bubbling with 100% O_2_ for 45 min. Two further digests were carried out using 400 U/mL collagenase (CLS II, Biochrom AG, Berlin, Germany) for 45 min. The cell suspension was filtered through 200 µm gauze to remove undigested tissue, and the myocytes were pelleted by gentle centrifugation (400 rpm). The pellet was thereafter resuspended in preoxygenated Krebs–Henseleit medium followed by resuspension of the cardiomyocytes in buffer with a stepwise increase in calcium.

Primary human umbilical vein endothelial cells (HUVECs) were isolated with collagenase IV as described previously by us [26]. The harvested cells were cultured in PromoCell^®^ endothelial cell basal medium supplemented with 10% (vol/vol) FCS, 0.4% (vol/vol) endothelial growth supplement with heparin, 0.1 ng/mL human EGF, 1.0 μg/mL hydrocortisone, 1 ng/mL human bFGF, and 2% (*v*/*v*) penicillin/streptomycin in humidified atmosphere at 37 °C, 5% CO_2_, and used at passage 1.

### 2.8. Detection of ROS Formation

For detection of reactive oxygen species (ROS) formation, RV or LV ARVCs were plated on black 12-wells and loaded with dichlorodihydrofluorescein diacetate (DCFH-DA) dye to measure intracellular ROS release, with CellROX^®^ Green to measure nuclear/mitochondrial ROS or with CellROX^®^ Deep Red (both ThermoFisher Scientific) to measure cytoplasmic ROS according to the manufacturer’s instructions. RV microvascular ECs were seeded on 96 wells at a density of 10,000 cells/well (passage 1). All cells were incubated with the according dye at 37 °C for 30 min, washed with PBS, and analyzed on a Fluostar Optima microplate fluorometer (BMG Laboratories (Ortenberg, Germany); DCFH-DA excitation 495 nm; emission 530 nm, CellROX^®^ Deep Red: excitation 640 nm; emission 665 nm, CellROX^®^ Green: excitation 485 nm; emission 520 nm) at 37 °C every 5 min.

### 2.9. Measurement of Apoptosis

Caspase 3 activity was detected with the luminometric Caspase-Glo^®^ 3/7 Assay (Promega GmbH, Mannheim, Germany) on a GloMax™ 96 Plate luminometer.

### 2.10. H9C2 Cardiomyoblast Culture and siRNA Transfection

The H9C2 rat cardiomyoblast cell line, which has many features of cardiomyocytes and has previously been shown to undergo hypertrophy in response to angiotensin II or endothelin-1 [27], was obtained from the American Type Culture Collection. H9C2 were maintained in DMEM supplemented with 10% FCS and 1% penicillin/streptomycin under an atmosphere of 5% CO_2_ in air at 37 °C. Twenty-four hours before transfection, H9C2 cells were trypsinized and transferred to 6-well plates (5 × 10^5^ cells/well). Adiponectin receptor-1 (AdipoR1), AdipoR2, T-cadherin, calreticulin, gC1qR (globular C1q receptor), sirtuin-3 (SIRT3), alpha1-AMPK, alpha2-AMPK, and control siRNA (FlexiTube siRNA, Qiagen, Hilden, Germany) oligonucleotides were transfected to the cells at a concentration of 0.5 nmol/L with Lipofectamine^®^ RNAiMAX (ThermoFisher Scientific). The sequences (Appendix A
Appendix A) had been digitally searched and no similarity to other genes was found in current databases. The control cells were transfected with control siRNA oligonucleotides with no known target in mammalian genomes. Forty-eight hours after siRNA transfection and after serum starvation for 24 h, H9C2 cells were treated with the indicated substances in serum-free medium.

### 2.11. RNA Isolation, RT-PCR and qPCR

Total RNA was isolated from tissue by guanidine thiocyanate/cesium chloride centrifugation. Total RNA from ARVCs, microvascular ECs and cardiac FBs was isolated using TriFast (Peqlab, Erlangen, Germany) according to the manufacturer’s instructions. Prior to cDNA synthesis, integrity and quality of the RNA was confirmed by gel electrophoresis and the concentration was determined by measuring UV absorption. Reverse transcription of RNA samples (500 ng total RNA) was carried out for 30 min at 42 °C using the SuperScript™ III First-Strand cDNA Synthesis Kit (ThermoFisher Scientific). RT-PCR was performed for rat AdipoR1, AdipoR2, T-Cadherin, thioredoxin-1 (Trx1), Trx2, superoxide dismutase-1 (SOD1), SOD2, glutathione peroxidase-1 (GPX1), GPX2, NADPH oxidase-2 (Nox2), Nox4, p22 phox, p47 phox, CTRP9, and 18S rRNA. Amplification products were subjected to electrophoresis through 1.5% agarose gels, stained with GelRed (VWR) and visualized with the Fusion FX7 imaging system (Peqlab). PCR products were excised from the gel, purified and directly sequenced. Real-time PCR and data analysis were performed using the Mx3000P Multiplex Quantitative PCR System (Stratagene, San Diego, CA, USA) as described previously [28]. Each assay was performed in duplicate and validation of the PCR runs was assessed by evaluation of the melting curve of the PCR products (primer sequences in Appendix A
Appendix A) and quantified with the ΔΔC_T_-method. Threshold cycles (C_T_) of target genes were normalized to the housekeeping gene *18S rRNA* (18S rRNA Control kit, Yakima Yellow^®^-Eclipse^®^ Dark Quencher, Eurogentec, Liege, Belgium). The resulting ΔC_T_ values were compared to sham animals (seven weeks after surgery) or untreated control cells and relative mRNA expression was calculated by R = 2^–ΔΔCT^.

### 2.12. Subcellular Fractionation and Western Blotting

Subcellular fractionation of cytoplasmic, membrane and nuclear fractions was carried out using the Subcellular Protein Fractionation Kit for Cultured Cells (Pierce), following the manufacturer’s instructions. The mitochondrial fraction was isolated with a buffer containing 250 mmol/L sucrose, 20 mmol/L HEPES (pH 7.5), 10 mmol/L KCl, 1.5 mmol/L MgCl_2_, 1 mmol/L EDTA, 1 mmol/L EGTA, 1 mmol/L DTT, and protease inhibitor cocktail (Sigma). Fractional samples were used for analysis of the localization of AdipoR1, AdipoR2, T-cadherin, calreticulin and gC1qR within cardiomyocytes. The cytosolic fraction was also utilized for the detection of cytochrome C. For all other Western blot analyses, ARVCs were homogenized in a buffer containing 50 mmol/L Tris HCl, 150 mmol/L NaCl, 5 mmol/L EDTA, 0.1% SDS, 1% sodiumdeoxycholate, and protease and phosphatase inhibitor cocktails (Sigma). Fifteen micrograms of protein were loaded on SDS-PAGE gels and transferred to nitrocellulose membranes. Following blocking, filters were incubated with antibodies directed against, phospho-AMPK (Thr172), alpha1-AMPK, alpha2-AMPK, alpha-AMPK, phospho-p44/42 MAPK, p44/42 MAPK, phospho-Akt (Thr308), Akt, phospho-p38MAPK, p38MAPK, acetyl-coenzyme A carboxylase (ACC), phospho-ACC, cytochrome C (all Cell Signaling Europe B.V., Frankfurt am Main, Germany), Trx1 (LabFrontier, Dongan-gu, Korea), SOD2 (Upstate/Millipore, Burlington, MA, USA), CTRP9 (BioTrend Chemikalien GmbH, Köln, Germany), T-cadherin (Abcam, Cambridge, UK), AdipoR1, AdipoR2 (both IBL International GmbH, Hamburg, Germany), calreticulin, gC1qR (both Abcam), FLAG, myc (both Cell Signaling), Caspase 9 (Cell signaling #9508), and Glyceraldehyde-3-phosphate dehydrogenase (GAPDH, Abcam). After incubation with peroxidase-conjugated secondary antibody, blots were subjected to the enhanced chemiluminescent detection method with the Fusion FX7 imaging system (Peqlab).

### 2.13. Co-Immunoprecipitation

The full-length cDNA of rat AdipoR1, AdipoR2, T-cadherin, calreticulin, and gC1qR was cloned into pENTR™/D-TOPO^®^ using the pENTR™ Directional TOPO cloning kit (ThermoFisher Scientific) and recombined into pcDNA3.1/nV5-DEST™ afterwards. An N-terminal myc-epitope tag was introduced into the AdipoR1 clone and an N-terminal FLAG-epitope tag was introduced into the other clones. For immunoprecipitation (IP), H9C2 myoblast were seeded on 10 cm plates and transfected with JetPRIME^®^ transfection reagent (Peqlab) according to the manufacturer’s instructions. 48 h after transfection, cells were solubilized in lysis buffer (50 mmol/L Tris, pH 7.5, 0.5% deoxycholic acid, 0.5% Triton X-100, 10 mmol/L EDTA, including protease and phosphatase inhibitors) and lysates clarified by centrifugation. Two-hundred micrograms of each cell lysate were used for immunoprecipitation. A mouse monoclonal myc antibody covalently bound to protein A/G agarose beads with the crosslinker dimethyl pimelimidate (DMP) or a mouse monoclonal Flag antibody covalently bound to magnetic Dynabeads™ (ThermoFisher Scientific) was used for precipitation. The lysate-beads/antibody conjugate mixture was incubated at 4 °C under rotary agitation overnight, washed five times to remove non-specific binding, and eluted afterward. Antibodies against FLAG or myc were used for detection respectively. IgG1 was used as an isotype control and an unrelated FLAG- or myc-tagged protein was used as a negative control in these IP experiments.

In order to stabilize protein-protein interactions, chemical crosslinking with the crosslinking reagent dimethyl-dithiobispropionimidate (DTBP) was performed in H9C2 cells overexpressing myc-AdipoR1. Samples were washed to remove excess crosslinker and lysates were prepared as described above and subjected to immunoprecipitation. Immune complexes were eluted by boiling in SDS sample buffer and reduced with dithiothreitol (DTT) to cleave DTBP.

### 2.14. Pathway Reporter Array

The Cignal 45 reporter array (QIAGEN) was used to identify signaling pathways that may be activated or repressed by CTRP9. A reverse transfection with jetPEI™ transfection reagent (Peqlab) was performed in H9C2 cardiomyoblasts (10,000 cells per well) according to the manufacturer’s instructions. In addition to the transcription factors included in the Cignal 45 reporter array, we also tested the activation of Forkhead box O3 (Foxo3a). A pGL3 Foxo3a reporter plasmid was purchased from Addgene (plasmid #1789, provided by M. Greenberg) and used to transfect H9C2 cells in a 96well format. To confirm the array data, pGL3 reporter plasmids containing an antioxidant response element (ARE) binding site (5′GTG ACA AAG CAC CCG TGA CAA AGC ACC CGT GAC AAA GCA CCC GTG ACA AAG CAC CCG TGA CAA AGC ACC CGT GAC AAA GCA-3′), nuclear factor kappa-light-chain-enhancer of activated B cells (NFKB) binding sites (AGT TGA GGG AAT TTC CGG GAA TTT CCG GGA CTT TCC GGG ACT TTC CAG GC) or activator protein 1 (AP-1) sites (TGA GTC AGT GAG TCA CTG ACT CAC TGA CTC ATG AGT CAG CTG ACT AAT GAC TAA TGA CTA ATG ACT AAT GAC TAA) as well as pGL3 basic nuclear factor of activated T-cells (NFAT; Addgene #17870 provided by J. Crabtree) and pGL3 myocyte enhancer factor-2 (MEF2; Addgene # 32967 provided by R. Prywes) were utilized. 24 h after transfection, medium was changed and CTRP9 added. Luciferase activity was measured using the Dual Luciferase Assay system (Promega) on a GloMax™ Luminometer (Promega) 48 h after transfection. Firefly luciferase served as the experimental reporter and Renilla luciferase as the normalizing reporter.

### 2.15. Reporter Genes and Reporter Gene Assays

A genomic DNA clone encoding the rat *SOD2* promoter region or the mouse *Trx1* promoter region was obtained by polymerase chain reaction (SOD2 promoter rat forward 5′- CCC TTA CCA AGC CCA GTC AT -3′, SOD2 promoter rat reverse 5′- TGC TGA GGC GCC CAC GA -3′; Trx1 promoter mouse forward 5′- ATT CCT CAG CCC TTC AAA AA -3′, Trx1 promoter mouse reverse 5′- CAC CAT TTT GGC TGT TGC-3′), amplifying a 1008 bp SOD2 fragment and a 1430 bp Trx1 fragment upstream of the ATG. The PCR products were gel-purified, cloned into pCR II TOPO (ThermoFisher Scientific) and sequenced. The *SOD2* promoter contains various consensus cis DNA elements such as a Foxo3a binding sites [29] and a nuclear factor erythroid 2-related factor 2 (Nrf2) binding site/antioxidant response element (ARE) [30]. The *Trx1* promoter also contains Foxo3 binding sites [31]. Site-directed mutagenesis of the Foxo3a or ARE binding sites was performed using the QuikChange^®^ Site-Directed Mutagenesis Kit (Stratagene). All products were confirmed by sequencing. Original and mutated promoter fragments were subcloned into pGL3 Enhancer (Promega). The promoter-firefly luciferase fusion DNA (1 µg) and 0.1 µg of pRL-TK control vector (Renilla luciferase) were introduced to H9C2 cardiomyoblasts with Lipofectamine^®^ Plus™ reagent (ThermoFisher Scientific). Twenty-four hours after transfection, cells were stimulated with CTRP9 for another 24 h. Luciferase activities were determined using the Dual-Glo^®^ Luciferase Assay System (Promega) on a GloMax™ Luminometer (Promega) with dual injectors.

### 2.16. Statistical Analysis

All data are presented as mean ± SEM. Statistical analyses were performed with SigmaStat 3.5 software (Systat Software GmbH, Erkrath, Germany). Data were analyzed for normal distribution (Shapiro-Wilk test) and variance (Levene test) and subsequently analyzed using student *t* test or ANOVA with post hoc analysis as appropriate. *p* values of < 0.05 were considered statistically significant.

## 3. Results

### 3.1. Two Clearly Distinct Stages of RV Disease are Reached

PAB resulted in a significant increase in RV weight seven weeks after surgery as a sign of RV hypertrophy which was further amplified 22 weeks after surgery (Figure 1A). LV weight was not altered at any of these time points, but total heart weight was also increased 22 weeks after PAB surgery. Liver weight increased 22 weeks after PAB surgery, suggesting hepatic congestion as a sign of RV failure (Figure 1A). Echocardiographic analyses demonstrated a significant deterioration in RV systolic function 22 weeks after PAB while LV function was not affected (Figure 1B). Plasma BNP, a clinically relevant marker of cardiac hypertrophy and failure, was increased seven weeks after surgery and further amplified 22 weeks after surgery (Figure 1C). All these parameters suggest that a stage of compensatory hypertrophy without impairment in RV function was reached seven weeks after surgery while 22 weeks after surgery clear signs of impaired RV function were detectable (decompensated stage). As shown in the Appendix A
Appendix A, lung weight was not different between the groups, thus corroborating the exclusive RV impairment and arguing against a contribution of the LV to the increased BNP plasma levels.

### 3.2. RV Failure is Associated with Altered Cardiomyocyte CTRP9 Expression

RV tissue at the compensated stage showed an increased expression of CTRP9 mRNA and protein while CTRP9 mRNA and protein were strongly decreased at the decompensated stage (Figure 2A). In order to define the cellular source of CTRP9 within the rat heart, we compared CTRP9 mRNA in cardiac cells. ARVCs and ECs isolated from adult rat hearts showed the highest CTRP9 mRNA expression, while adult cardiac FBs expressed no CTRP9 mRNA (Figure 2B). Interestingly, RV cardiomyocytes at the decompensated stage showed a significant decline in CTRP9 mRNA and protein expression compared to those from sham rats 22 weeks after surgery (Figure 2C). No significant difference was observed in the according RV ECs (Appendix A
Appendix A) or in LV cardiomyocytes (Figure 2C). In addition, similar analyses for the identification of the cellular CTRP9 source were also performed in human patient samples. Our analyses suggested that CTPR9 mRNA and protein expression are highest in human cardiomyocytes and significantly lower in HUVECs or human vascular samples (aorta or mammary artery). The expressional level of CTRP9 in human cardiomyocytes reached similar levels as in human epicardial adipose tissue or in human blood samples (Figure 2D). This suggests that the cardiomyocyte itself is a major source of cardiac CTRP9. In addition, cardiac cells may experience exposure to high local CTRP9 levels through paracrine secretion from epicardial adipocytes.

Next, we investigated potential mediators of cardiac CTRP9 expression. Although metformin and insulin have been described to induce adiponectin [32,33], they did not induce CTRP9 mRNA (Figure 2E) in ARVCs. Incubation of ARVCs with the adrenergic agonists norepinephrine; isoproterenol or phenylephrine, however, resulted in the induction of CTRP9 mRNA. This inducing effect was strongly inhibited by the beta-adrenoceptor antagonist propranolol but only mildly influenced by the alpha1-adrenoceptor antagonist prazosin (Figure 2E).

### 3.3. CTRP9 Mediates Anti-Oxidative Effects

RV cardiomyocytes at the decompensated stage showed a significant increase in cellular ROS compared to those from sham rats 22 weeks after surgery (Figure 3A). This increase in ROS may have contributed to the increased apoptotic activation (increased caspase 3 activity, increased cleaved caspase 9) in RV cardiomyocytes at the decompensated stage (Figure 3A,B). No significant difference was observed in the according RV ECs (Figure 3A). LV cardiomyocytes isolated from PAB animals at the decompensated stage also demonstrated a mild increase in ROS production (Figure 3A). This phenomenon may be related to the fact that cardiomyocytes from the interventricular septum, which are affected by RV dysfunction, are included within the LV cardiomyocytes due to technical reasons (isolation procedure). No change in caspase 3 activity was observed in RV ECs or LV cardiomyocytes (Figure 3A). Similarly, Western blot analyses showed increased amounts of cleaved caspase 9 mainly at the stage of RV decompensation in RV cardiomyocytes but not in ECs or LV cardiomyocytes (Figure 3B).

Next we investigated, whether CTRP9 has an impact on the increased ROS production known to be induced in ARVCs by pro-hypertrophic mediators such as phenylephrine, endothelin-1 or angiotensin II. Previously, signaling pathway activation by CTRP9 (2–4 μg/mL) had been investigated in C2C12 myotubes or adipose-derived mesenchymal stem cells by others [10,34]. Accordingly, we tested different concentrations of recombinant CTRP9 in adult rat cardiomyocytes (Appendix A
Appendix A) and analyzed the CTRP9-induced activation of AMPK and its downstream target ACC. All further experiments were thereafter performed with 4 µg/mL CTRP9. Measurements of ROS production in ARVCs revealed that phenylephrine, angiotensin II and endothelin-1 induced an increased ROS release from mitochondrial and from cytosolic sources which was significantly blunted after 24 h pre-incubation with CTRP9 (Figure 3C). Accordingly, we observed increased caspase 9 activation in response to phenylephrine, angiotensin II or endothelin which was partially prevented by CTRP9 pre-incubation (Figure 3D). In line with this, the cytosolic fraction isolated from such cardiomyocytes showed increased cytochrome c release which was largely attenuated by CTRP9 pre-incubation (Figure 3D). Short-term pre-incubation (1 h) with CTRP9 did not modify the phenylephrine, angiotensin II and endothelin-1 induced increase in ROS release from mitochondrial and cytosolic sources (Figure 3E), suggesting that CTRP9 itself does not act as an antioxidant.

### 3.4. Transcriptional Regulation of Enzymes Involved in ROS Production or ROS Defense by CTRP9

#### 3.4.1. Influence of CTRP9 on the Expression of Enzymes Involved in ROS Production or Defense in ARVCs

As shown above (Figure 3B), pre-incubation with CTRP9 for 24h blunted the ROS release induced by various pro-hypertrophic agents. We hypothesized that CTRP9 may mediate these anti-oxidant effects through transcriptional or translational mechanisms. Accordingly, changes in the expression of ROS producing enzymes or enzymes involved in ROS defense may be involved in the anti-oxidative effects of CTRP9. Indeed, CTRP9 treatment resulted in a significant, AMPK-dependent increase in the mRNA expression of catalase, mitochondrial SOD2 and cytosolic Trx1 in ARVCs (Figure 4A). Other anti-oxidative enzymes or cofactors such as Trx2, SOD1, or GPX1 and GPX2 were not altered (Figure 4A). Analyses of the mRNA expression of potential ROS producing enzymes such as Nox2, Nox4, p47 phox, and p22 phox revealed no significant differences between controls and CTRP9-treated ARVCs (Figure 4B).

Next, we also tested the anti-oxidant effects of CTRP9 in the setting of an increased endogenous ROS production in addition to effects on the receptor-mediated increased ROS production by phenylephrine, endothelin-1 or angiotensin II. For this purpose, we utilized paraquat, which acts as a redox cycling compound and has been shown to increase the endogenous superoxide production mainly at the mitochondria [35]. Indeed, pre-treatment of ARVCs with CTRP9 for 24 h was also effective in reducing paraquat-induced ROS. When antioxidant defense was compromised by inhibiting Trx-1 with PX-12, the protective effects of CTRP9 against paraquat were partially abolished (Figure 4C). Since the commonly used SOD blocker diethyldithiocarbamate does not inhibit SOD2, we used CsA, which has recently been shown to inhibit SOD2 activity through tyrosine 34 nitration [36]. CsA was sufficient to totally block the antioxidant effects of CTRP9 in the presence of paraquat (Figure 4C).

#### 3.4.2. Influence of CTRP9 on the Expression of Anti-Oxidant Enzymes and Potentially Involved Transcription Factors in Cardiomyoblasts

Similar to ARVCs, H9C2 cardiomyoblasts, treated with CTRP9 (4 µg/mL) for 24 h, also demonstrated a significant increase in SOD2 mRNA, which was accompanied by a similar increase in protein expression and promoter activation (Figure 5A). Our initial screens for transcription factors involved in the effects of CTRP9 revealed a strong activation of Foxo3a, ARE, and NFkB (array data, not shown). In addition, some transcription factors known to be involved in pro-hypertrophic effects such as AP-1 or NFAT showed a weaker activation after CTRP9 treatment (array data, not shown). In additional experiments utilizing individual luciferase reporter plasmids for those transcription factors identified in the initial screens, we were able to confirm the activation of Foxo3a, NFkB and ARE, while no significant change in the activation of AP-1 or NFAT was observed after CTRP9 treatment (Appendix A
Appendix A). The SOD2 promoter contains an Nrf2 binding site/anti-oxidant response element and Foxo3a binding sites. Mutation of the ARE or the Foxo3a site within the SOD2 promoter completely abolished the reporter gene activation in response to CTRP9 (Figure 5A). The AMPK activator metformin, which has previously been shown to induce the expression of SOD2 mRNA in endothelial cells [37], was used as a positive control in these analyses. Unlike the experiments performed with CTRP9, the response to metformin was not totally abolished after mutation of the ARE site (Figure 5A). This suggests that in addition to AMPK, which is activated by metformin and by CTRP9, AMPK-independent signaling pathways may be involved in mediating these metformin effects (Figure 5A). CTRP9 also induced Trx1 mRNA and protein expression and resulted in increased Trx1 promoter activation in H9C2 cells (Figure 5B). The Trx1 promoter has also been described to contain Foxo3a binding sites. The mutant Trx1 luciferase reporter (ΔFoxo3a) demonstrated a loss of activation after CTRP9 treatment (Figure 5B). Unlike in ARVCs, catalase expression was not altered in H9C2 cardiomyoblasts in response to CTRP9 (not shown).

Previously, Foxo3a-dependent activation of anti-oxidant genes was shown to require SIRT3 [38,39]. In our experiments, SIRT3 knockdown resulted in a diminished CTRP9-induced AMPK activation, SOD2 protein expression and Trx1 protein expression in H9C2 cardiomyoblasts, suggesting the involvement of SIRT3 in CTRP9-induced anti-oxidant effects (Figure 5C). The strongest effects on basal and CTRP9-induced expression were observed for SOD2 (Figure 5C). SIRT3 knockdown also eliminated the anti-oxidant effects of CTRP9 after phenylephrine treatment (Figure 5C).

### 3.5. Signaling Pathways Involved in Anti-Oxidant CTRP9 Effects

Recombinant CTRP9 (4 µg/mL) resulted in a strong activation of AMPK, p44/42 MAPK and p38 MAPK in ARVCs (Figure 6A) and in H9C2 cells (not shown) within 10 min. While the activation of p44/42 MAPK and p38 MAPK was only transient, AMPK activation was long-lasting (24 h) and resulted also in a sustained activation of the AMPK downstream target ACC for up to 24 h (not shown). Akt activation, which has been described to occur in response to some CTRPs in other cell types, was not different between the groups (Figure 6A). No differences in signaling pathway activation were observed when ARVCs were stimulated with prokaryotic CTRP9 or with eukaryotic CTRP9 (not shown). The inhibitory effects of CTRP9 on phenylephrine-induced ROS production were abolished after pre-incubation with the AMPK inhibitor Ara A, but unaffected after inhibition of the MEK/ERK pathway with UO 126 or p38MAPK inhibition with SB202190, suggesting a major role of AMPK in this process (Figure 6B). In accordance with these results, alpha1-AMPK or alpha2-AMPK knockdown in H9C2 cells abolished the anti-oxidative effects of CTRP9 pre-treatment (Figure 6C). Alpha1-AMPK knockdown also resulted in a significant increase in basal ROS, while a similar effect on basal ROS production was not observed after alpha2-AMPK knockdown (Figure 6C). Similar results were obtained for mitochondrial ROS (CellROX^®^ Green) or cytosolic ROS (CellROX^®^ Red) as shown in Figure 6C.

### 3.6. Role of Adiponectin Receptors in CTRP9-Mediated Signal Transduction

Although AdipoR1 has been suggested to be involved in CTRP9 signaling [12,40], AdipoR1 knockdown did not attenuate the CTRP9-induced anti-oxidant effects (Figure 6D). Similarly, exclusive AdipoR2 or T-cadherin knockdown did not modify the anti-oxidant CTRP9 effects. Finally, a concomitant 80% knockdown of both adiponectin receptors resulted in significantly blunted, but not totally abolished anti-oxidant CTRP9 effects in response to phenylephrine stimulation (Figure 6E), suggesting the existence of additional CTRP9 receptors. Another surface molecule, gC1qR (globular C1q receptor = globular C1q binding protein (C1qbp)), was reported to recognize the globular head of C1q-related proteins and to show a high affinity to adiponectin [41]. However, gC1qR knockdown did not modify the anti-oxidant effects of CTRP9 (Figure 6F). In addition to the adiponectin receptors, calreticulin was suggested to be involved in adiponectin signal transduction [42] and may, thus, also be involved in CTRP9-induced signal transduction. Indeed, calreticulin knockdown resulted in a stronger ROS production compared to mock cells under basal conditions and after phenylephrine stimulation, which was no longer blunted by CTRP9 (Figure 6F).

Next, we investigated whether AdipoR1/2 and T-cadherin directly interact with calreticulin. Subcellular fractionation revealed strongest expression of AdipoR1, AdipoR2 and T-cadherin in the membrane fraction of ARVCs (Figure 7A). Calreticulin and gC1qR were detected in the cytosol, nucleus and mitochondria in addition to the membrane fraction (Figure 7A), suggesting the potential for an interaction with adiponectin receptors. Immunoprecipitation showed an interaction of AdipoR1 with AdipoR2 and T-cadherin, but no direct interaction with calreticulin or gC1qR (Figure 7B). Similar results were obtained for AdipoR2 (not shown). To identify a potential AdipoR1-calreticulin interaction, we also used a crosslinker in order to unveil transient protein-protein interactions. Although changes in the migration pattern were detected on AdipoR1 Western blots in comparison to experiments without crosslinker, calreticulin could not be identified as a potential direct interactor of the adiponectin receptors or of CTRP9 with this technique (not shown).

## 4. Discussion

The present study shows that CTRP9, which is most likely released by the cardiomyocyte itself, induces an anti-oxidative protection through SIRT3-, AMPK-, and Foxo3a-dependent transcriptional activation of Trx1 and SOD2 in cardiomyocytes (Figure 8). These protective CTRP9 effects appear to be preserved at the stage of compensatory RV hypertrophy but they are lost at the stage of RV failure.

Although AdipoR1 has been reported to be involved in CTRP9 signaling by others [12,40], AdipoR1 knockdown in H9C2 cells did not attenuate the CTRP9-induced anti-oxidant effects in our study. Only a concomitant knockdown of both adiponectin receptors, which are almost equally expressed in H9C2 cardiomyoblasts, attenuated the CTRP9-induced anti-oxidant effects. This suggests that not only AdipoR1 but also AdipoR2 is able to mediate the CTRP9 response. Others have suggested that CTRP receptors may be distinct from adiponectin receptors, since overexpression of T-cadherin in multiple cell types did not result in enhanced binding of CTRP9 [10], and phosphorylation of AMPK induced by CTRP5 is unaffected by AdipoR1 or AdipoR2 depletion [43]. AdipoR1 is mainly involved in AMPK activation, but AdipoR2 is associated with the activation of the PPARα pathway [44]. Similar to adiponectin, most CTRPs exist in low molecular weight (LMW) and high molecular weight (HMW) isoforms of 12–18 units [4,45]. We utilized mainly CTRP9 derived from *E. coli*, which does not form HMW isoforms. However, no differences in signaling pathway activation were observed when ARVCs were stimulated with prokaryotic CTRP9 or with eukaryotic CTRP9 in our study. HMW isoforms of adiponectin have been shown to be able to activate AMPK more potently than LMW isoforms, and T-cadherin preferentially binds the HMW forms of adiponectin [1,2]. Similar analyses have not been performed for the CTRPs so far. Furthermore, CTRP9 [10] and other CTRPs [4,45] form homomers and heteromers with other CTRPs or with adiponectin, which may also influence receptor affinity and downstream signaling.

In addition, calreticulin was suggested to be involved in adiponectin signal transduction [42] and is therefore also a potential CTRP9 receptor. Calreticulin is a multifunctional calcium-buffering chaperone of the endoplasmic reticulum (ER). It is involved in protein folding, maturation, and trafficking outside the ER and, thus, participates in the synthesis of a variety of molecules, such as surface receptors or ion channels. Adiponectin binding to calreticulin has been demonstrated on the cell surface of macrophages and neonatal cardiomyocytes by flow cytometry and immunoprecipitation [42]. Our knockdown experiments provide evidence that calreticulin may also be involved in CTRP9 signaling, although a direct interaction of calreticulin with the adiponectin receptors or with CTRP9 was not observed. Recently, it has been shown by others that CTRP9 binds to calrecticulin and is involved in its anti-apoptotic effects through PKA-dependent mechanisms in a model of ischemia/reperfusion injury [14]. This comprehensive study performed in mice and rats suggested that AdipoR1, AMPK, MAPK, and Akt are not involved in the protective CTRP9 effects [14]. Unlike our study, those in vitro studies were performed in neonatal cardiomyocytes and the in vivo experiments investigated LV pathologies [14]. Furthermore, no difference in PKA activation in response to CTRP9 was observed in our study in ARVCs.

Alpha1 or alpha2-AMPK knockdown in H9C2 cells abolished the anti-oxidative effects of CTRP9 pre-treatment, suggesting that both isoforms are required. However, it has been shown by others that CTRP9 remains cardioprotective in mice with cardiomyocyte-specific overexpression of a dominant negative alpha2-AMPK [13]. Previously, Jorgensen et al. demonstrated an upregulation of the alpha 1 isoform in skeletal muscle of alpha2-AMPK KO mice [46]. A similar phenomenon can be envisioned in mice with dominant negative alpha2-AMPK, where alpha1-AMPK may partially compensate for the loss of alpha2-AMPK. In our experiments, alpha1-AMPK knockdown resulted in a significant increase in basal ROS in H9C2 cells, which may be related to the expressional pattern of alpha-AMPK isoforms observed in H9C2 cells. While ARVCs and the adult heart demonstrated a preferential expression of the alpha2 isoform, H9C2 cells showed a higher expression of the alpha 1 isoform (not shown).

Cardiac expression of CTRP9 exceeds adiponectin by more than 100-fold [7]. Although adiponectin is also produced by cardiomyocytes [47], CTRP9 has emerged as a major cardioprotective mediator in situations with adiponectin deficit such as obesity, ageing or heart failure [6,28,48]. Our study demonstrates a significant increase in CTRP9 expression in the RV at the stage of compensatory hypertrophy, which could represent a local, cardiac-specific upregulation to compensate for the systemic adiponectin deficit in heart disease. Our study shows that CTRP9 expression is as high in human epicardial adipose tissue as in human cardiomyocytes. This might represent an important functional aspect in terms of a potential paracrine secretion of CTRP9 from epicardial adipocytes and its impact on various cardiac cells. Patients with diabetes or metabolic syndrome exhibit an increased inflammatory burden in pericoronary adipose tissue, as well as increased cardiac oxidative stress [16,49]. Furthermore, circulating CTRP9 as well as adiponectin levels are significantly decreased in patients with coronary artery disease, diabetes or obesity [16,50,51]. Novel therapeutic approaches in cardiovascular diseases may thus also target adipose tissue since the adipose-derived CTRP9 as well as adiponectin have been shown to mediate strong anti-inflammatory and anti-oxidative vascular effects [50,52,53].

AMPK-induced activation of Foxo3 leads to an increased transcription of various anti-oxidative enzymes including thioredoxin, SOD2 and catalase [54], which involves nuclear translocation of Foxo3 [31]. Although nuclear translocation was not investigated, the involvement of Foxo3 is clearly shown in our study. However, Foxo3 alone is not sufficient to antagonize increased ROS during neuroendocrine activation but requires SIRT3 [38]. SIRT3 deacetylates and activates the AMPK upstream kinase LKB1, resulting in an increased AMPK activation [39]. Furthermore, increased activity of SIRT3 protects cardiomyocytes from oxidative stress-mediated cell death by increased expression of the anti-oxidative enzymes SOD2 and catalase [38]. Our study suggests that SIRT3 is required for the CTRP9-induced and AMPK-mediated anti-oxidative effects in RV cardiomyocytes through transcriptional activation of SOD2 and Trx1. Similarly, anti-apoptotic and cell survival promoting effects of CTRP9 on mesenchymal stem cells as well as adipose-derived stem cells have been described to involve increased SOD2, SOD3, peroxiredoxin-1, and heme oxygenase-1 expression via Nrf2 and ERK activation [34,55]. No increased AMPK or Akt activation was observed in these studies, suggesting cell-type specific differences in CTRP9 signal transduction [34]. Others have shown that the SIRT3-dependent activation of anti-oxidative Foxo3a targets Ras activation and represses the activity of transcription factors such as GATA4 and NFAT involved in the development of cardiac hypertrophy [38]. Therefore, the CTRP9-induced minor decrease in NFAT activation in cardiomyocytes observed in our study requires further investigations. In addition, it remains to be elucidated through which mechanisms the CTRP9 effects on AMPK and SIRT3 are mediated.

CTRP9 was shown to attenuate adverse cardiac remodeling through inhibition of cardiomyocyte apoptosis and fibrosis in a mouse model of acute myocardial infarction [13]. In addition to AMPK, activation of PKA was involved in cardioprotective actions of CTRP9 during postischemic remodeling [13]. The observed inhibitory in vivo effects of CTRP9 could also involve anti-oxidative effects on cardiomyocytes, which were not investigated in the study by Sun et al. [13]. In addition to an increased anti-oxidative defense, a lower ROS production could play a role in the cardioprotective effects of CTRP9. Although this has indeed been described to occur in an in vivo model of cardiac I/R [7], the expression of ROS producing enzyme such as NADPH oxidases was not influenced by CTRP9 application in our study. Finally, unlike most other investigations on cardiac effects of CTRP9, another study showed that CTRP9 induces deterioration in LV function [8]. The authors show that endothelial cell-derived CTRP9 triggers cardiomyocyte hypertrophy and promotes cardiac failure in a mouse TAC-model [8]. In their model, no difference in the activation of AMPK, which has been shown to be involved in the protective CTRP9 effects in the present study but also in many others, was observed. Instead, they observed an enhanced ERK5 activation leading to phosphorylation and activation of the pro-hypertrophic transcription factor GATA-4 [8]. Cell-type and ventricle-specific differences in signal transduction may have contributed to the contrasting results on cardiac CTRP9 effects.

Unlike previous studies, our investigations focus on the cardioprotective effects of CTRP9 in a model of stage-dependent RV pressure overload. In addition, alterations in cardiomyocytes and endothelial cells derived from the overloaded RV as well as alterations in LV and RV cardiomyocytes are compared. Our set of data shows that CTRP9 activates an anti-oxidative response through an adiponectin receptor/calreticulin, AMPK, and Foxo3a-dependent transcriptional activation of Trx1 and SOD2 in cardiomyocytes, which antagonizes the ROS-inducing effects of various pro-hypertrophic agents (Figure 8). However, there are limitations in this study that could be addressed in future research. First, we utilized mainly CTRP9 derived from *E. coli*, which does not form HMW isoforms and may, therefore, mediate less potent effects via AMPK and T-cadherin [1,2]. Secondly, major parts of the study were performed in H9C2 cardiomyoblast which differ from adult cardiomyocytes in gene expression [56] including the expressional pattern of the alpha-AMPK isoforms (own data). Finally, future studies should investigate in more detail the specific role of Foxo3 and SIRT3 in mediating the CTRP9 effects in cardiomyocytes.

## 5. Conclusions

Our results provide additional understanding of the mechanisms underlying the cardioprotective effects of CTRP9, and extend our knowledge to RV pathologies for the first time. The data also suggests that the loss of CTRP9 at the stage of RV failure may have detrimental consequences. Therefore, CTRP9 may be valuable for mitigation or even prevention of pathological RV remodeling.

## Figures and Tables

**Figure 1 cells-09-01229-f001:**
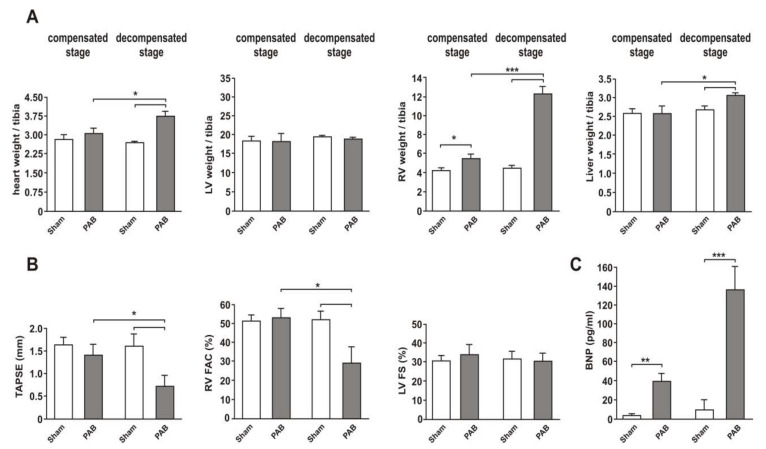
Animal characteristics in the two-stage PAB model. (**A**) Changes in heart weight (weight in g *100/tibia in mm), LV weight and RV weight (weight in mg/tibia in mm) as well as liver weight (weight in g *10/tibia in mm) all normalized to tibia length in sham animals and PAB animals seven weeks after surgery (compensated stage) or 22 weeks after surgery (decompensated stage). (**B**) Changes in tricuspid annular plane systolic excursion (TAPSE), RV fractional area change (RV FAC), and LV fractional shortening (LV FS) in sham animals and PAB animals seven or 22 weeks after surgery. (**C**) Plasma BNP levels in sham animals and PAB animals seven or 22 weeks after surgery. All data are mean ± SEM, *n* = 8 animals per group, * *p* < 0.05; ** *p* < 0.01; *** *p* < 0.001.

**Figure 2 cells-09-01229-f002:**
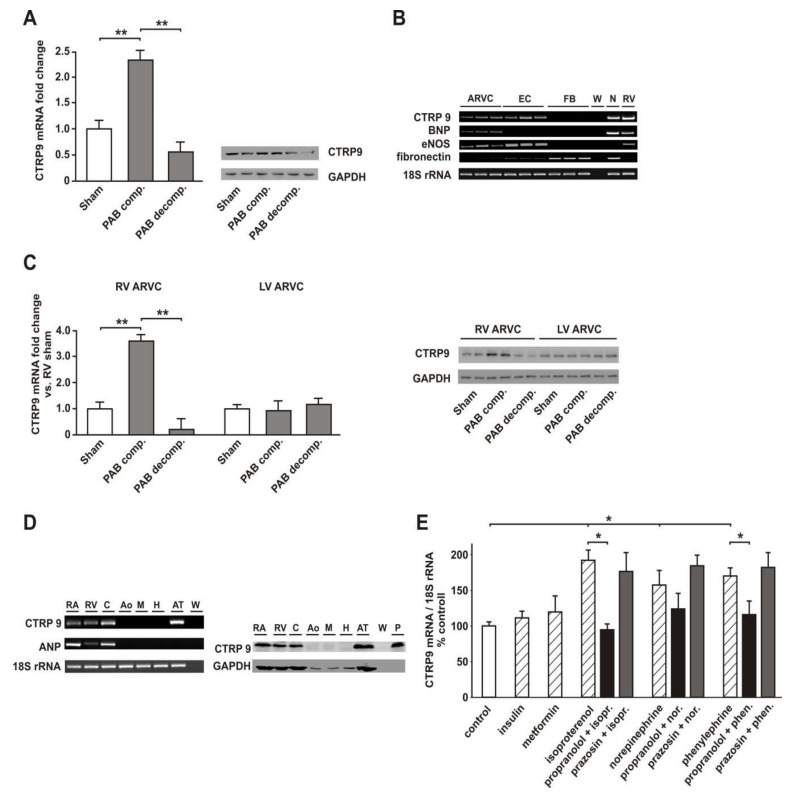
CTRP9 expression in rat and human samples. (**A**) Changes in CTRP9 expression in RV tissue from sham animals and PAB animals at the stage of compensated hypertrophy (PAB comp.) or RV decompensation (PAB decomp.) as determined by real-time PCR and Western blotting. Representative CTRP9 blots from RV tissue with GAPDH as loading control are shown. n = 8 animals per group. (**B**) Expressional pattern of CTRP9 mRNA was analyzed in RV ARVCs, RV cardiac microvascular ECs, RV FBs, neonatal rat cardiomyocytes (N), and rat RV tissue (RV). Brain natriuretic peptide (BNP) served as a cardiomyocyte-specific marker. Endothelial NO synthase (eNOS) shows highest expression in endothelial cells, while fibronectin is strongly expressed in fibroblasts. 18S ribosomal RNA was used as a house-keeping gene. Water (W) served as a no template control. n = cells from six different animals per group. (**C**) Changes in CTRP9 mRNA and protein expression in RV and LV ARVCs from sham animals and PAB animals at the stage of RV compensation or RV decompensation as determined by real-time PCR and Western blotting. *n* = cells from five different animals per group. (**D**) CTRP9 mRNA and protein expression was analyzed in human right atrial tissue (RA), right ventricular tissue (RV), human RA cardiomyocytes (C), human aorta (Ao), human mammary artery (M), human umbilical vein endothelial cells (H), human epicardial adipose tissue (AT) and in human plasma (P; only protein expression). 18S ribosomal RNA (PCR) or GAPDH protein (Western blot) were used as a house-keeping gene. W (water, no-template control), ANP (atrial natriuretic peptide). Samples from 5–10 patients were utilized, depending on the availability of the different tissue biopsies. (**E**) Changes in the mRNA expression in ARVCs after 24 h stimulation with insulin (100 nmol/L), metformin (0.5 mmol/L), isoproterenol (5 µmol/L), norepinephrine (1 µmol/L) or phenylephrine (10 µmol/L) as determined by quantitative real-time PCR. Pre-incubation (30 min) with propranolol (1 µmol/L) or prazosin (1 µmol/L) was performed as indicated. *n* = cells from 11–20 different animals per group. All data are mean ± SEM, * *p* < 0.05; ** *p* < 0.01.

**Figure 3 cells-09-01229-f003:**
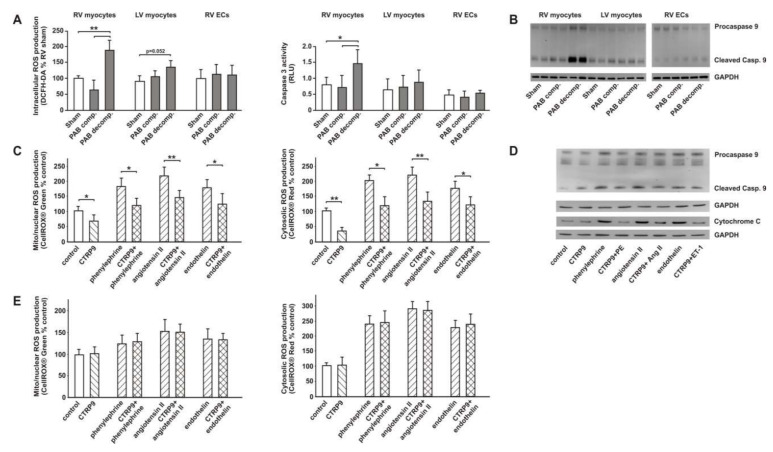
ROS release and apoptosis in RV cardiomyocytes and impact of CTRP9 on ROS production during pro-oxidant stimulation. (**A**) Cellular ROS release was analyzed with DCFH-DA in freshly isolated RV ARVCs and LV ARVCs 6 h after plating and in RV ECs 24 h after plating. As an indicator of apoptosis, caspase 3 activity (relative light units × 100.000) was analyzed in these cells as well. *n* = cells from 6 different animals per group. (**B**) Caspase 9 activation as determined by Western Blotting. Representative blots from RV ARVCs, LV ARVCs or ECs from sham and PAB animals with GAPDH as loading control are shown. *n* = 6 animals per group. (**C**) Mitochondrial and cytosolic ROS release in ARVCs, pre-incubated for 24 h with CTRP9 (4 µg/mL) and afterwards stimulated with phenylephrine (10 µmol/L), angiotensin II (100 nmol/L) or endothelin-1 (100 nmol/L) for 1 h. *n* = cells from 16 different animals per group. (**D**) Caspase 9 activation (total cell lysate) and cytochrome C release (cytosolic fraction) as determined by Western Blotting. Representative blots from ARVCs, pre-incubated for 24 h with CTRP9 (4 µg/mL) and afterwards stimulated with phenylephrine (PE, 10 µmol/L), angiotensin II (Ang II, 100 nmol/L), or endothelin-1 (ET-1, 100 nmol/L) for 1 h. *n* = cells from five different animals per group. GAPDH served as loading control. (**E**) Mitochondrial and cytosolic ROS in ARVCs, pre-incubated for 1 h with CTRP9 (4 µg/mL) and afterwards stimulated with the pro-oxidant substances phenylephrine (10 µmol/L), angiotensin II (100 nmol/L) or endothelin-1 (100 nmol/L) for 1 h. *n* = cells from 16 different animals per group. All data are mean±SEM, * *p* < 0.05; ** *p* < 0.01.

**Figure 4 cells-09-01229-f004:**
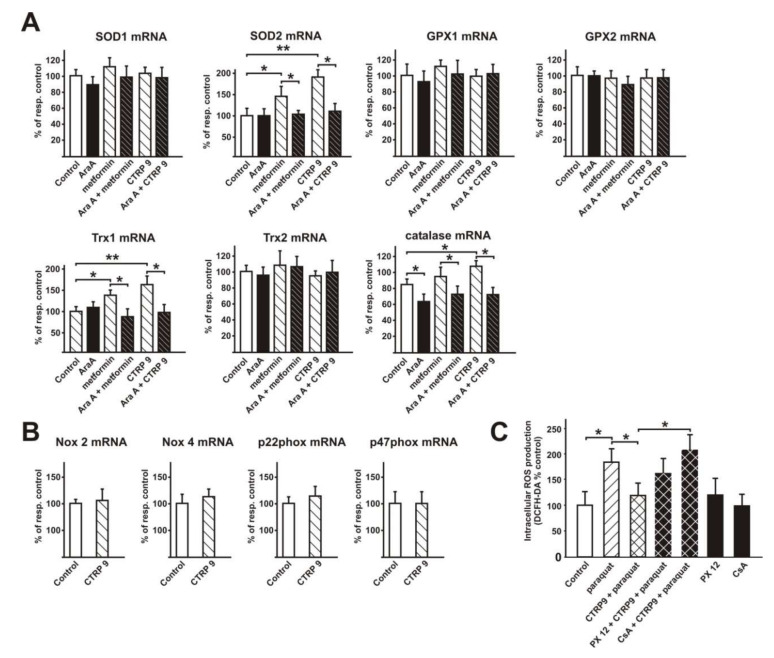
Influence of CTRP9 on the expression of anti-oxidative enzymes in ARVCs. (**A**) mRNA expression of cytosolic SOD1, mitochondrial SOD2, GPX1, GPX2, cytosolic Trx1, mitochondrial Trx2 and catalase was analyzed in ARVCs after stimulation with CTRP9 (4 µg/mL, 24 h). Metformin (0.5 mmol/L) served as a positive control. *n* = cells from eight different animals per group. (**B**) mRNA expression of Nox-2, Nox-4, p22 phox and p47 phox was analyzed in ARVCs after stimulation with CTRP9 (4 µg/mL, 24 h). *n* = cells from eight different animals per group. (**C**) Changes in cellular ROS release in response to 50 µmol/L paraquat in ARVCs after 24 h pre-incubation with CTRP9 (4 µg/mL). Trx1 was inhibited with PX 12 (10 µmol/L) and MnSOD was inhibited with CSA (1 µg/mL). *n* = cells from five different animals per group. All data are mean ± SEM, * *p* < 0.05; ** *p* < 0.01.

**Figure 5 cells-09-01229-f005:**
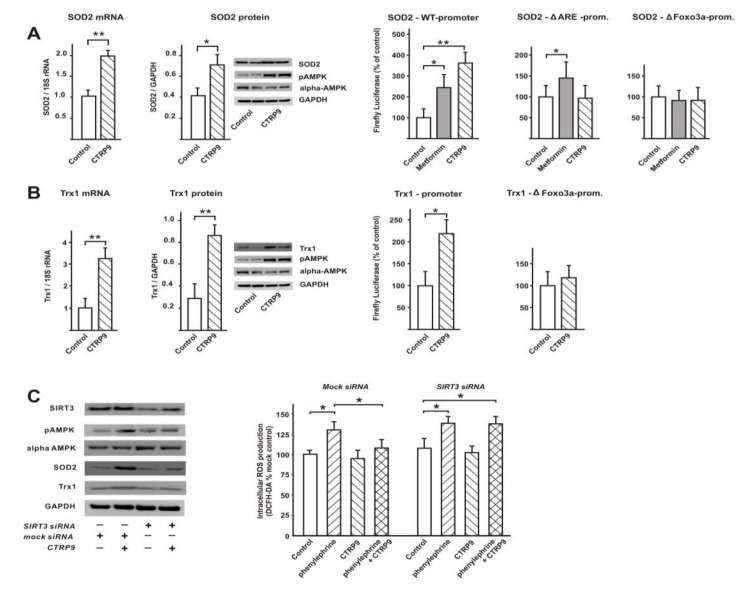
Influence of CTRP9 on anti-oxidative enzymes in cardiomyoblasts. (**A**) Influence of CTRP9 treatment (24 h) on mRNA expression, protein expression and promoter activation of SOD2 in H9C2 cells. Representative Western blots are shown. Mutation of the Nrf2/ARE or Foxo3a site abolished the promoter activation. Metformin served as a positive control. Data from five independent experiments. (**B**) Influence of CTRP9 treatment (24 h) on mRNA expression, protein expression and promoter activation Trx1 in H9C2 cells. Representative Western blots are shown. Mutation of the Foxo3a site abolished the promoter activation. Data from five independent experiments. (**C**) SIRT3 knockdown by siRNA in H9C2 cardiomyoblasts, mock transfected cells served as controls. AMPK activation and protein expression of SOD2 or Trx1 was analyzed after treatment with CTRP9 (4 µg/mL, 24 h). Cellular ROS release was analyzed with DCFH-DA in the accordingly treated cells after 24 h phenylephrine incubation. Data from five independent experiments. All data are mean ± SEM, * *p* < 0.05; ** *p* < 0.01.

**Figure 6 cells-09-01229-f006:**
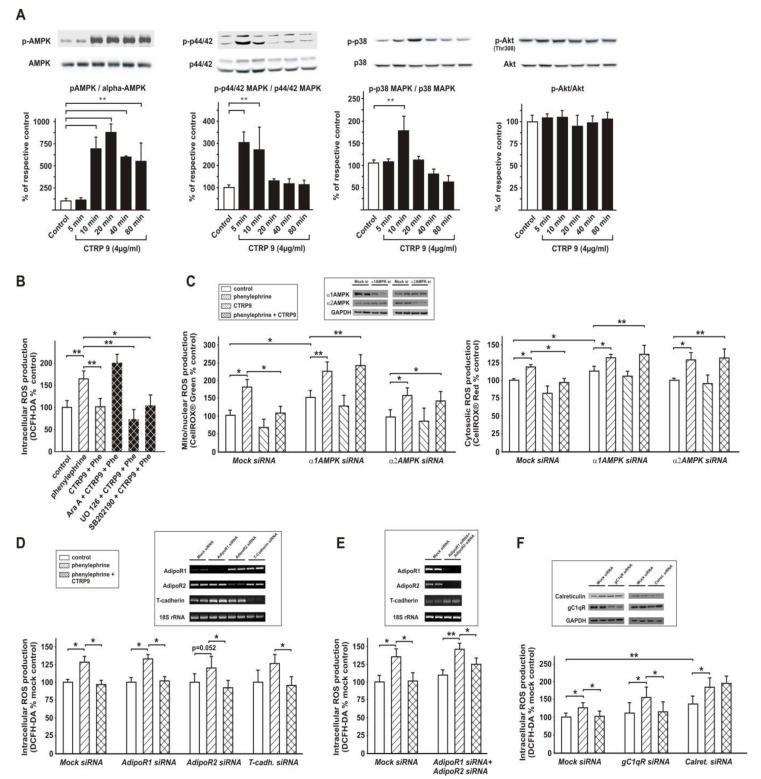
Signaling pathways involved in CTRP9 effects in cardiomyocytes and cardiomyoblasts. (**A**) Densitometry and representative Western blots, showing the time-dependent influence of CTRP9 (4 µg/mL) on the activation of AMPK, p44/42 MAPK, p38 MAPK and Akt in ARVCs. *n* = cells from 13–14 different animals per group. (**B**) Influence of AMPK inhibition with Ara A (500 µmol/L), p44/42 MAPK inhibition with U0126 (10 µmol/L) or p38 MAPK inhibition with SB 202190 (5 µmol/L) on CTRP9-mediated anti-oxidative effects as deduced from changes of DCFH-DA fluorescence in ARVCs in response to phenylephrine (10 µmol/L). Inhibitors were applied 30 min before CTRP9 (4 µg/mL, 24 h). *n* = cells from five different animals per group. (**C**): Mitochondrial and cytosolic ROS release in H9C2 cardiomyoblasts after alpha1 or alpha2-AMPK knockdown by siRNA. Mock transfected cells served as controls. 24 h after transfection, cells were pre-incubated with CTRP9 (4 µg/mL) for 24 h and then treated with phenylephrine. Knockdown efficiency (protein) is shown. Data from 5 independent experiments. (**D**) Influence of exclusive knockdown of AdipoR1, AdipoR2, or T-cadherin on CTRP9-mediated anti-oxidative effects as deduced from changes of DCFH-DA fluorescence in H9C2 cardiomyoblasts. Twenty-four hours after transfection, cells were pre-incubated with CTRP9 (4 µg/mL) for 24 h and then treated with phenylephrine. Knockdown efficiency (mRNA) is shown. Data from five independent experiments. (**E**) Influence of combined knockdown of AdipoR1 and AdipoR2 on CTRP9-mediated anti-oxidative effects as described in D. Data from five independent experiments. (**F**) Influence of knockdown of calreticulin or gC1qR on CTRP9-mediated anti-oxidative effects as described in D. Knockdown efficiency (protein) is shown. Data from five independent experiments. All data are mean ± SEM, * *p* < 0.05; ** *p* < 0.01.

**Figure 7 cells-09-01229-f007:**
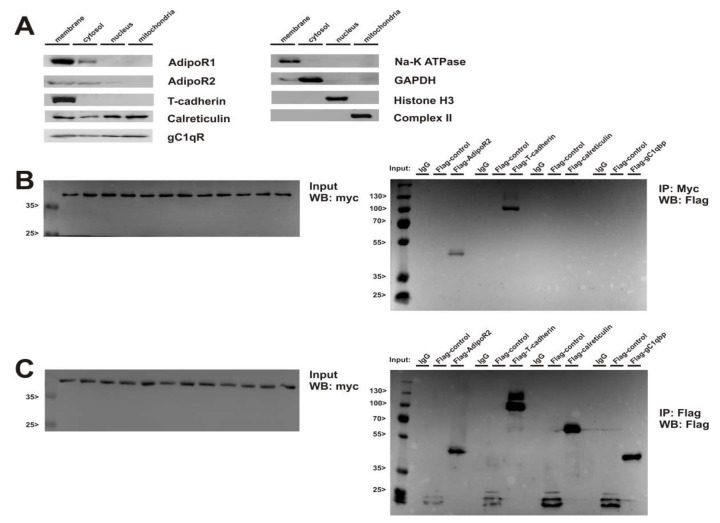
Cellular localization of potential CTRP9 receptors and their interaction. (**A**) Representative Western Blots from cellular fractions of ARVCs. The presence of the potential CTRP9 receptors AdipoR1, AdipoR2, T-cadherin, calreticulin, and gC1qR was analyzed in these fractions. Antibodies detecting Na-K-ATPase, GAPDH, histone H3 or cytochrome C demonstrate purity of the respective fractions. (**B**,**C**) H9C2 cardiomyoblasts transiently expressing myc-AdipoR1 or FLAG-AdipoR2, FLAG-T-cadherin, FLAG-calreticulin, and FLAG-gC1qR, respectively, were lysed and tagged receptors were immunoprecipitated. The presence of AdipoR1 was detected by immunoblotting for the myc-epitope. Co-immunoprecipitated AdipoR2 and T-cadherin were detected by immunoblotting for the FLAG-epitope (B). Immunoprecipitation and immunoblotting for the FLAG epitope demonstrates binding of the FLAG-epitope of AdipoR2, T-cadherin, calreticulin, gC1qR, and an unrelated 20 kDa protein, which was used as an internal control (C). No signal was detected, when IgG was used for the IP. Data shown are from one of 3-8 independent experiments.

**Figure 8 cells-09-01229-f008:**
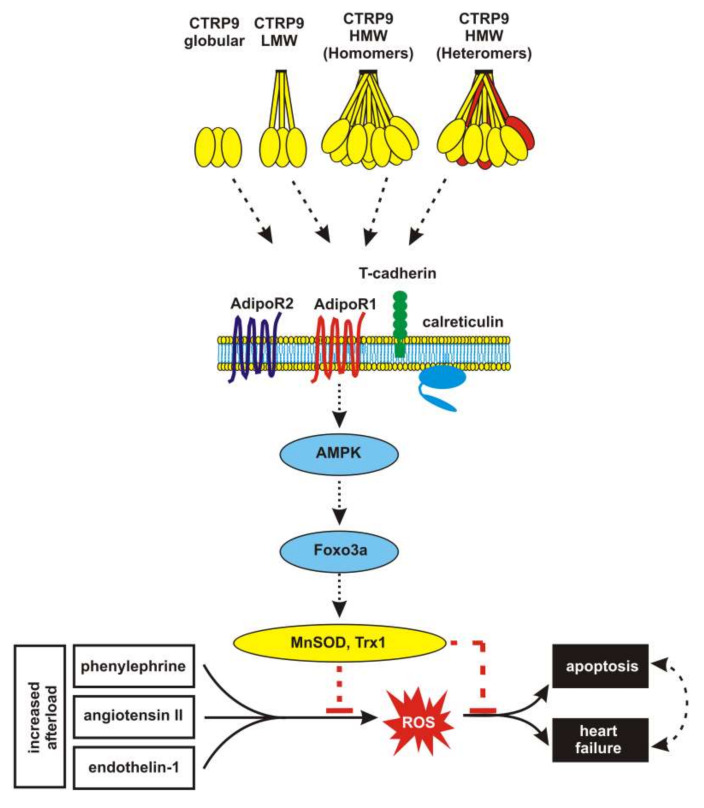
Summary of mechanisms involved in CTRP9-induced anti-oxidative effects. CTRP9 exists as globular, LMW and HMW isoform and can also form heteromers with adiponectin or other CTRPs. CTRP9 activates AMPK in cardiomyocytes. AMPK activation depends on AdipoR1 and AdipoR2 but also involves the co-receptor T-cadherin and the multifunctional protein calreticulin. AMPK-induced activation of Foxo3 leads to an increased transcription of anti-oxidative enzymes including Trx1 and SOD2, resulting in reduced cellular ROS. Thus, CTRP9 is able to antagonize the ROS-involving and apoptosis-inducing effects of phenylephrine, angiotensin II, or endothelin.

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
