# Peer review of "CTRP9 Mediates Protective Effects in Cardiomyocytes via AMPK- and Adiponectin Receptor-Mediated Induction of Anti-Oxidant Response"

_cells, 2020, doi:10.3390/cells9051229_

Round 1

Reviewer 1 Report

Dear Editor,

I carefully read the manuscript by Niemann et al., which is interesting and balanced in its parts.

Some comments for the Authors:

  • In my opinion the Keywords are too numerous.
  • English language needs to be revised throughout the manuscript in order to correct the typos
  • Figure 2B and 2D have characters written too small to be read. Please, improve the quality of the figure before resubmission.
  • Several references are dated and should be replaced with newer articles.
  • In the discussion, Authors should clearly state the strengths and limitations of their study.

Author Response

Reply to reviewer #1

We are grateful for the constructive evaluation of our paper. We incorporated the comments as follows below by the point to point reply.

Reviewer comment:

In my opinion the Keywords are too numerous.

Response:

We have significantly reduced the number of keywords to:

heart failure, ROS; AMPK; C1q family, cardiomyocyte

Reviewer comment:

English language needs to be revised throughout the manuscript in order to correct the typos

Response:

We have carefully revised the manuscript and corrected the typos.

Reviewer comment:

Figure 2B and 2D have characters written too small to be read. Please, improve the quality of the figure before resubmission.

Response:

We have increased the characters in Fig. 2B and 2D as suggested.

Reviewer comment:

Several references are dated and should be replaced with newer articles.

Response:

We have replaced some references by more recent publications with some exception such as methodical papers or initial descriptions of gene products or gene knockouts.

Reviewer comment:

In the discussion, Authors should clearly state the strengths and limitations of their study.

Response:

A new paragraph on strengths and limitations of the study has been included in the discussion of the revised manuscripts as follows:

“Unlike previous studies, our investigations focus on the cardioprotective effects of CTRP9 in a model of stage-dependent RV pressure overload. In addition, alterations in cardiomyocytes and endothelial cells derived from the overloaded RV as well as alterations in LV and RV cardiomyocytes are compared. Our set of data shows that CTRP9 activates an anti-oxidative response through an adiponectin receptor / calreticulin, AMPK and Foxo3a-dependent transcriptional activation of Trx1 and SOD2 in cardiomyocytes, which antagonizes the ROS-inducing effects of various pro-hypertrophic agents (Fig. 8). However, there are limitations in this study that could be addressed in future research. First, we utilized mainly CTRP9 derived from E. coli, which does not form HMW isoforms and may therefore mediate less potent effects via AMPK and T-cadherin [1,2]. Secondly, major parts of the study were performed in H9C2 cardiomyoblast which differ from adult cardiomyocytes in gene expression including the expressional pattern of the alpha-AMPK isoforms. Finally, future studies should investigate in more detail the specific role of Foxo3 and SIRT3 in mediating the CTRP9 effects in cardiomyocytes.”

References:

  1. Fang, H.; Judd, R.L. Adiponectin Regulation and Function. Comprehensive Physiology 2018, 8, 1031-1063, doi:10.1002/cphy.c170046.
  2. Wang, Z.V.; Scherer, P.E. Adiponectin, the past two decades. Journal of molecular cell biology 2016, 8, 93-100, doi:10.1093/jmcb/mjw011.

Reviewer 2 Report

The issues addressed in the manuscript are within the scope of the journal. The results and the topic of the study are interesting.

Detailed comments:

Abstract: Please explain the abbreviations

Introduction: Explain the abbreviations when used for the first time throughout the text

Author Response

Reply to reviewer #2

Many thanks for the thorough evaluation of our paper. We incorporated the comments as follows:

Reviewer comment:

Abstract: Please explain the abbreviations

Response:

We added an explanation for all abbreviations. However, the text had to be slightly modified in order to stay within the word limit (200 words).

Reviewer comment:

Introduction: Explain the abbreviations when used for the first time throughout the text

Response:

We have carefully checked the manuscript for all abbreviations. They should now all be explained at their first appearance in the text. We have labeled the according changes throughout the revised manuscript with red font.

Reviewer 3 Report

Dear editor,

About the manuscript entitled “Type of manuscript: Article

Title: CTRP9 mediates protective effects in cardiomyocytes via AMPK- and

adiponectin receptor-mediated induction of anti-oxidant response”, enclosed my suggestions:

I have several major concerns; the results are not convincing:

  1. In the manuscript is not described why CTRP9 is used at 4 μg/ml, therefore in line 124, it should be necessary to add specific references in which it is reported and explained why CTRP9 is used at this concentration;
  2. Line 164-168: The line space is different than the other lines;
  3. Figure 2:
  4. (B) it should be modified the upper part of figure, it is confusing. The names of different cells line could be writing in one line or in another clear way;
  5. (C) In figure 2B, you showed me that CTRP9 cellular source are ventricular cardiomyocytes and endothelial cells and in figure 2C you demonstrated that in decompensated stage there is no significant difference in the RV ECs. It should be demonstrated what happened in RV ECs in PAB compensated. In order to understand in this stage what are the cells that produce CTRP9;
  6. Figure 3: (A-B) To be sure that ROS increased apoptotic activation, it should be necessary to demonstrated the activation of other apoptotic markers as caspase 9 and cytochrome c;
  7. Figure 4: (A) Under the bars are not reported the name of samples;
  8. Figure 5: (A) Metformin is used as a positive control. Why does metformin not work well in SOD2 - Δ Fox3a-promoer? If metformin binds this site it is necessary to explain it.

About adiponectin and its implication in cardiac damage, please report the suggested points and reference:

  1. As first, remark the importance of adiponectin in the restenosis and overall new PCI in patients with insulin resistance and normal glucose tolerance (Cardiovasc Diabetol. 2019 Mar 4;18(1):24. doi: 10.1186/s12933-019-0826-0). Thus adiponectin could be evaluated by peripheral blood of patients with ischemic heart disease , and it could be used to predict worse prognosis (Cardiovasc Diabetol. 2019 Mar 4;18(1):24. doi: 10.1186/s12933-019-0826-0). Please discuss this point and this reference.
  2. Please stress the concept of over inflammation/oxidative stress effects on endothelial function and Major Adverse Cardiac Events ( Cardiovasc Diabetol. 2019 Sep 30;18(1):126. doi: 10.1186/s12933-019-0931-0; Curr Pharm Des. 2020 Feb 13. doi: 10.2174/1381612826666200213123029). This is relevant.
  3. About endothelial dysfunction and over inflammation/oxidative stress in diabetes, I would suggest you to underline the effects induced also by pre-diabetes status on Coronary Endothelial Dysfunction in Patients with Stable Angina and Non obstructive Coronary Artery Stenosis (Diabetes Care. 2019 Oct;42(10):1946-1955. doi: 10.2337/dc18-2356), and to the opposite the cadioprotective pathways induced by cardiomyocyte-derived exosomal epigenetic regulators, that mediate post-ischemic myofibroblast activation both in vitro and ex vivo models (ESC Heart Fail. 2020 Feb;7(1):284-288. doi: 10.1002/ehf2.12584). However, these conditions are induced by over inflammation and shift gradually into fibrotic responses with post-ischemic activation. Please discuss these points and these references in your manuscript.
  4. Again, it looks relevant to remember the progression of cardiovascular disease of diabetics in heart failure (Front Physiol. 2019 Mar 19;10:253. doi: 10.3389/fphys.2019.00253), and its refractory to advanced and interventional therapies with the consequent worse prognosis (Cardiovasc Diabetol. 2018 Oct 22;17(1):137. doi: 10.1186/s12933-018-0778-9; Cardiovasc Diabetol. 2017 Jun 9;16(1):75. doi: 10.1186/s12933-017-0554-2). Please introduce these informations in your manuscript.

Author Response

Reply to reviewer #3

We would like to thank for the thorough evaluation of our paper. We incorporated the comments as follows below by the point to point reply.

Reviewer comment:

In the manuscript is not described why CTRP9 is used at 4 μg/ml, therefore in line 124, it should be necessary to add specific references in which it is reported and explained why CTRP9 is used at this concentration;

Response: We have added the following comments and references in the manuscript as well as a new supplementary figure.

“Previously, signaling pathway activation by CTRP9 (2-4 μg/ml) had been investigated in C2C12 myotubes or adipose-derived mesenchymal stem cells by others [3,4]. Accordingly, we tested different concentrations of recombinant CTRP9 in adult rat cardiomyocytes (Suppl. Fig. 3) and analyzed the CTRP9-induced activation of AMPK and its downstream target ACC. All further experiments were thereafter performed with 4µg/ml CTRP9.”

Reviewer comment:

Line 164-168: The line space is different than the other lines;

Response:

We have corrected the according difference in spacing.

Reviewer comment:

Figure 2:(B) it should be modified the upper part of figure, it is confusing. The names of different cells line could be writing in one line or in another clear way;

Response:

We have modified the labeling of Figure 2B in order to improve comprehensibility.

Reviewer comment:

In figure 2B, you showed me that CTRP9 cellular source are ventricular cardiomyocytes and endothelial cells and in figure 2C you demonstrated that in decompensated stage there is no significant difference in the RV ECs. It should be demonstrated what happened in RV ECs in PAB compensated. In order to understand in this stage what are the cells that produce CTRP9;

Response:

We added data on CTRP9 mRNA and protein expression in RV endothelial cells at the compensated stage. For this purpose new Western blots had to be performed. These data were included in the Supplementary Fig. 2 of the revised manuscript.

Reviewer comment:

Figure 3: (A-B) To be sure that ROS increased apoptotic activation, it should be necessary to demonstrated the activation of other apoptotic markers as caspase 9 and cytochrome c;

Response:

First, we have analyzed caspase 9 activation in existing total cell lysates from adult rat cardiomyocytes and endothelial cells of sham and PAB animals. The according data are presented in Fig. 3B of the revised manuscript. In addition, we conducted additional experiments in adult rat cardiomyocytes and investigated caspase 9 activation and cytochrome C release into cytosol  as shown in Fig. 3D of the revised manuscript.

The results obtained from these analyses show increased caspase 9 activation (detection of increased amounts of cleaved caspase 9) mainly at the stage of RV decompensation in RV cardiomyocytes but not in ECs or LV cardiomyocytes (Fig. 3B). Cytochrome C could not be investigated in these samples since no cytosolic fraction had been isolated at the time of preparation. It was impossible within 10 days to provide such samples due to the time course of the animal model (7 and 22 weeks after PAB surgery).

We also analyzed caspase 9 activation in cardiomyocytes treated with phenylephrine, angiotensin II or endothelin after pre-incubation with CTRP9 (Fig. 3D). The results obtained from these analyses show an increased caspase 9 activation in response to phenylephrine, angiotensin II or endothelin which is partially prevented by CTRP9 pre-incubation. In line with this, the cytosolic fraction isolated from such cardiomyocytes shows increased cytochrome c release which is prevented by CTRP9 pre-incubation (Fig. 3D). However, there was not sufficient material available from ECs to perform these analyses. Furthermore, the focus of our analyses are cardiomyocytes. Therefore, we are convinced that the according EC data are dispensable.

Reviewer comment:

Figure 4: (A) Under the bars are not reported the name of samples;

Response:

We have added the according labeling.

Reviewer comment:

Figure 5: (A) Metformin is used as a positive control. Why does metformin not work well in SOD2 - Δ Fox3a-promoter? If metformin binds this site it is necessary to explain it.

Response:

Indeed, metformin has been described to induce the expression of SOD2 mRNA in endothelial cells through an AMPK-dependent mechanism [5]. It was therefore used as a positive control in our experiments. Activation of the SOD2 promoter was not investigated in that study [5]. Further details on signaling pathways activated by metformin resulting in the activation of the SOD2 promoter have not been investigated in detail by us. Recent investigations by others suggest a major contribution of epigenetic mechanisms in this process [6-8].

Our study shows that activation of the SOD2 promoter requires a functional ARE as well as Foxo3a binding site, since both mutations resulted in a significantly reduced luciferase activity in response to metformin and in response to CTRP9. It is likely that activation of AMPK, which is achieved by metformin and by CTRP9, and AMPK-dependent signaling pathways play a major role in this process.

Therefore, we have specified the reported effects in ECs and included an additional comment in the revised manuscript as follows:

“The AMPK activator metformin, which has previously been shown to induce the expression of SOD2 mRNA in endothelial cells [5], was used as a positive control in these analyses. Unlike the experiments performed with CTRP9, the response to metformin was not totally abolished after mutation of the ARE site (Fig. 5A). This suggests that in addition to AMPK, which is activated by metformin and by CTRP9, AMPK-independent signaling pathways may be involved in mediating these metformin effects (Fig. 5A).”

Reviewer comment:

About adiponectin and its implication in cardiac damage, please report the suggested points and reference:

As first, remark the importance of adiponectin in the restenosis and overall new PCI in patients with insulin resistance and normal glucose tolerance (Cardiovasc Diabetol. 2019 Mar 4;18(1):24. doi: 10.1186/s12933-019-0826-0). Thus adiponectin could be evaluated by peripheral blood of patients with ischemic heart disease , and it could be used to predict worse prognosis (Cardiovasc Diabetol. 2019 Mar 4;18(1):24. doi: 10.1186/s12933-019-0826-0). Please discuss this point and this reference.

Please stress the concept of over inflammation/oxidative stress effects on endothelial function and Major Adverse Cardiac Events ( Cardiovasc Diabetol. 2019 Sep 30;18(1):126. doi: 10.1186/s12933-019-0931-0; Curr Pharm Des. 2020 Feb 13. doi: 10.2174/1381612826666200213123029). This is relevant.

About endothelial dysfunction and over inflammation/oxidative stress in diabetes, I would suggest you to underline the effects induced also by pre-diabetes status on Coronary Endothelial Dysfunction in Patients with Stable Angina and Non obstructive Coronary Artery Stenosis (Diabetes Care. 2019 Oct;42(10):1946-1955. doi: 10.2337/dc18-2356), and to the opposite the cadioprotective pathways induced by cardiomyocyte-derived exosomal epigenetic regulators, that mediate post-ischemic myofibroblast activation both in vitro and ex vivo models (ESC Heart Fail. 2020 Feb;7(1):284-288. doi: 10.1002/ehf2.12584). However, these conditions are induced by over inflammation and shift gradually into fibrotic responses with post-ischemic activation. Please discuss these points and these references in your manuscript.

Again, it looks relevant to remember the progression of cardiovascular disease of diabetics in heart failure (Front Physiol. 2019 Mar 19;10:253. doi: 10.3389/fphys.2019.00253), and its refractory to advanced and interventional therapies with the consequent worse prognosis (Cardiovasc Diabetol. 2018 Oct 22;17(1):137. doi: 10.1186/s12933-018-0778-9; Cardiovasc Diabetol. 2017 Jun 9;16(1):75. doi: 10.1186/s12933-017-0554-2). Please introduce these informations in your manuscript.

Response:

We would like to thank the reviewer for pointing us to these additional aspects related to adiponectin and its implication on cardiac damage. We particularly appreciate his/her effort to emphasize the potential clinical relevance of our study.

The main topic of our study is the effect of the adiponectin paralog CTRP 9 on cardiomyocytes with a particular focus on cardioprotective mechanisms of CTRP9. Although CTRP9 is structurally related to adiponectin, there are clear differences between these two mediators in terms of origin, mode of action and signaling pathways.

The 8 articles suggested by the reviewer have a clear focus on clinical aspects related to the progression of cardiovascular diseases in diabetics. Similarly, there are also various publications discussing the association of CTRP 9 with arteriosclerosis, arterial stiffness, and inflammation in patients with type 2 diabetes and insulin resistance. However, we believe that this aspect of CTRP9 research may sidetrack the reader of our article from the main topic. In addition, some of the articles suggested by the reviewer do not seem to be within the primary focus of our article considering that there are limitations in word count, number of references etc. Therefore, we could only include the following two of the suggested articles as well as two CTRP9-related manuscripts in the introduction of our manuscript as follows:

“In addition, locally expressed mediators including adiponectin or the related CTRPs may represent novel biomarkers for the prediction of prognosis and progression of cardiovascular diseases [9-12].”

References:

  1. Fang, H.; Judd, R.L. Adiponectin Regulation and Function. Comprehensive Physiology 2018, 8, 1031-1063, doi:10.1002/cphy.c170046.
  2. Wang, Z.V.; Scherer, P.E. Adiponectin, the past two decades. Journal of molecular cell biology 2016, 8, 93-100, doi:10.1093/jmcb/mjw011.
  3. Wong, G.W.; Krawczyk, S.A.; Kitidis-Mitrokostas, C.; Ge, G.; Spooner, E.; Hug, C.; Gimeno, R.; Lodish, H.F. Identification and characterization of CTRP9, a novel secreted glycoprotein, from adipose tissue that reduces serum glucose in mice and forms heterotrimers with adiponectin. FASEB J 2009, 23, 241-258, doi:10.1096/fj.08-114991.
  4. Yan, W.; Guo, Y.; Tao, L.; Lau, W.B.; Gan, L.; Yan, Z.; Guo, R.; Gao, E.; Wong, G.W.; Koch, W.L., et al. C1q/Tumor Necrosis Factor-Related Protein-9 Regulates the Fate of Implanted Mesenchymal Stem Cells and Mobilizes Their Protective Effects Against Ischemic Heart Injury via Multiple Novel Signaling Pathways. Circulation 2017, 136, 2162-2177, doi:10.1161/CIRCULATIONAHA.117.029557.
  5. Kukidome, D.; Nishikawa, T.; Sonoda, K.; Imoto, K.; Fujisawa, K.; Yano, M.; Motoshima, H.; Taguchi, T.; Matsumura, T.; Araki, E. Activation of AMP-activated protein kinase reduces hyperglycemia-induced mitochondrial reactive oxygen species production and promotes mitochondrial biogenesis in human umbilical vein endothelial cells. Diabetes 2006, 55, 120-127, doi:10.2337/diabetes.55.01.06.db05-0943.
  6. Bridgeman, S.C.; Ellison, G.C.; Melton, P.E.; Newsholme, P.; Mamotte, C.D.S. Epigenetic effects of metformin: From molecular mechanisms to clinical implications. Diabetes, obesity & metabolism 2018, 20, 1553-1562, doi:10.1111/dom.13262.
  7. Elbere, I.; Silamikelis, I.; Ustinova, M.; Kalnina, I.; Zaharenko, L.; Peculis, R.; Konrade, I.; Ciuculete, D.M.; Zhukovsky, C.; Gudra, D., et al. Significantly altered peripheral blood cell DNA methylation profile as a result of immediate effect of metformin use in healthy individuals. Clinical epigenetics 2018, 10, 156, doi:10.1186/s13148-018-0593-x.
  8. Cuyas, E.; Fernandez-Arroyo, S.; Joven, J.; Menendez, J.A. Metformin targets histone acetylation in cancer-prone epithelial cells. Cell Cycle 2016, 15, 3355-3361, doi:10.1080/15384101.2016.1249547.
  9. Sardu, C.; Paolisso, G.; Marfella, R. Molecular mechanisms and therapeutic targets of inflammatory-related Cardiovascular diseases: from molecular mechanisms to therapeutic targets. Curr Pharm Des 2020, 10.2174/1381612826666200213123029, doi:10.2174/1381612826666200213123029.
  10. Sardu, C.; D'Onofrio, N.; Torella, M.; Portoghese, M.; Loreni, F.; Mureddu, S.; Signoriello, G.; Scisciola, L.; Barbieri, M.; Rizzo, M.R., et al. Pericoronary fat inflammation and Major Adverse Cardiac Events (MACE) in prediabetic patients with acute myocardial infarction: effects of metformin. Cardiovasc Diabetol 2019, 18, 126, doi:10.1186/s12933-019-0931-0.
  11. Gao, C.; Zhao, S.; Lian, K.; Mi, B.; Si, R.; Tan, Z.; Fu, F.; Wang, S.; Wang, R.; Ma, X., et al. C1q/TNF-related protein 3 (CTRP3) and 9 (CTRP9) concentrations are decreased in patients with heart failure and are associated with increased morbidity and mortality. BMC cardiovascular disorders 2019, 19, 139, doi:10.1186/s12872-019-1117-0.
  12. Wang, J.; Hang, T.; Cheng, X.M.; Li, D.M.; Zhang, Q.G.; Wang, L.J.; Peng, Y.P.; Gong, J.B. Associations of C1q/TNF-Related Protein-9 Levels in Serum and Epicardial Adipose Tissue with Coronary Atherosclerosis in Humans. BioMed research international 2015, 2015, 971683, doi:10.1155/2015/971683.

Round 2

Reviewer 3 Report

Dear author,

I see an improvement in the present manuscript.

On other hand, I want suggest you to report, as previously discussed:

1. the importance of adiponectin in the restenosis and overall new PCI in patients with insulin resistance and normal glucose tolerance (Cardiovasc Diabetol. 2019 Mar 4;18(1):24. doi: 10.1186/s12933-019-0826-0). From here, the message for readers could be that adiponectin could be evaluated by peripheral blood of patients with ischemic heart disease , and it could be used to predict worse prognosis (Cardiovasc Diabetol. 2019 Mar 4;18(1):24. doi: 10.1186/s12933-019-0826-0). Please discuss this point and this reference.

2. Please stress the concept of over inflammation/oxidative stress effects on endothelial function and Major Adverse Cardiac Events ( Cardiovasc Diabetol. 2019 Sep 30;18(1):126. doi: 10.1186/s12933-019-0931-0).

Author Response

Reply to reviewer #3

We would like to thank for the evaluation of our revised paper.

Reviewer comment:

I see an improvement in the present manuscript.

On other hand, I want suggest you to report, as previously discussed:

  1. the importance of adiponectin in the restenosis and overall new PCI in patients with insulin resistance and normal glucose tolerance (Cardiovasc Diabetol. 2019 Mar 4;18(1):24. doi: 10.1186/s12933-019-0826-0). From here, the message for readers could be that adiponectin could be evaluated by peripheral blood of patients with ischemic heart disease , and it could be used to predict worse prognosis (Cardiovasc Diabetol. 2019 Mar 4;18(1):24. doi: 10.1186/s12933-019-0826-0). Please discuss this point and this reference.
  2. Please stress the concept of over inflammation/oxidative stress effects on endothelial function and Major Adverse Cardiac Events (Cardiovasc Diabetol. 2019 Sep 30;18(1):126. doi: 10.1186/s12933-019-0931-0).

Response:

We have included the two additional suggested articles as follows. The changes are marked in blue font in the revised manuscript.

line 64-67:

“Elevated CTRP 3 levels for example have been related to the prevalence and severity of coronary artery disease, suggesting its potential as a novel biomarker [1]. Similarly, it has been shown in a single-center prospective longitudinal study that adiponectin could be utilized as a biomarker for the prediction of restenosis and worse prognosis in patients with coronary artery disease [2].”

line 651-660:

“Our study shows that CTRP9 expression is as high in human epicardial adipose tissue as in human cardiomyocytes. This might represent an important functional aspect in terms of a potential paracrine secretion of CTRP9 from epicardial adipocytes and its impact on various cardiac cells. Patients with diabetes or metabolic syndrome exhibit an increased inflammatory burden in pericoronary adipose tissue as well as increased cardiac oxidative stress [3,4]. Furthermore, circulating CTRP9 as well as adiponectin levels are significantly decreased in patients with coronary artery disease, diabetes or obesity [3,5,6]. Novel therapeutic approaches in cardiovascular diseases may thus also target adipose tissue since the adipose-derived CTRP9 as well as adiponectin have been shown to mediate strong anti-inflammatory and anti-oxidative vascular effects [5,7,8]”.

References:

  1. Wang, S.; Ling, Y.; Liang, W.; Shen, L. Association of serum C1q/TNF-related protein-3 (CTRP-3) in patients with coronary artery disease. BMC cardiovascular disorders 2017, 17, 210, doi:10.1186/s12872-017-0646-7.
  2. Sasso, F.C.; Pafundi, P.C.; Marfella, R.; Calabro, P.; Piscione, F.; Furbatto, F.; Esposito, G.; Galiero, R.; Gragnano, F.; Rinaldi, L., et al. Adiponectin and insulin resistance are related to restenosis and overall new PCI in subjects with normal glucose tolerance: the prospective AIRE Study. Cardiovasc Diabetol 2019, 18, 24, doi:10.1186/s12933-019-0826-0.
  3. Sardu, C.; D'Onofrio, N.; Torella, M.; Portoghese, M.; Loreni, F.; Mureddu, S.; Signoriello, G.; Scisciola, L.; Barbieri, M.; Rizzo, M.R., et al. Pericoronary fat inflammation and Major Adverse Cardiac Events (MACE) in prediabetic patients with acute myocardial infarction: effects of metformin. Cardiovasc Diabetol 2019, 18, 126, doi:10.1186/s12933-019-0931-0.
  4. Niemann, B.; Rohrbach, S.; Miller, M.R.; Newby, D.E.; Fuster, V.; Kovacic, J.C. Oxidative Stress and Cardiovascular Risk: Obesity, Diabetes, Smoking, and Pollution: Part 3 of a 3-Part Series. J Am Coll Cardiol 2017, 70, 230-251, doi:10.1016/j.jacc.2017.05.043.
  5. Yu, X.H.; Zhang, D.W.; Zheng, X.L.; Tang, C.K. C1q tumor necrosis factor-related protein 9 in atherosclerosis: Mechanistic insights and therapeutic potential. Atherosclerosis 2018, 276, 109-116, doi:10.1016/j.atherosclerosis.2018.07.022.
  6. Arita, Y.; Kihara, S.; Ouchi, N.; Takahashi, M.; Maeda, K.; Miyagawa, J.; Hotta, K.; Shimomura, I.; Nakamura, T.; Miyaoka, K., et al. Paradoxical decrease of an adipose-specific protein, adiponectin, in obesity. Biochem Biophys Res Commun 1999, 257, 79-83.
  7. Jung, C.H.; Lee, M.J.; Kang, Y.M.; Lee, Y.L.; Seol, S.M.; Yoon, H.K.; Kang, S.W.; Lee, W.J.; Park, J.Y. C1q/TNF-related protein-9 inhibits cytokine-induced vascular inflammation and leukocyte adhesiveness via AMP-activated protein kinase activation in endothelial cells. Mol Cell Endocrinol 2016, 419, 235-243, doi:10.1016/j.mce.2015.10.023.
  8. Okamoto, Y.; Kihara, S.; Ouchi, N.; Nishida, M.; Arita, Y.; Kumada, M.; Ohashi, K.; Sakai, N.; Shimomura, I.; Kobayashi, H., et al. Adiponectin reduces atherosclerosis in apolipoprotein E-deficient mice. Circulation 2002, 106, 2767-2770, doi:10.1161/01.cir.0000042707.50032.19.